



# Signature of the stratosphere-troposphere coupling on recent record-breaking Antarctic sea ice anomalies

Raúl R. Cordero[1,2], Sarah Feron[2]*, Alessandro Damiani[3], Pedro J. Llanillo[4], Jorge Carrasco[5], Alia L. Khan[6,7], Richard Bintanja[8,9], Zutao Ouyang[10], Gino Casassa[5]

[1]Universidad de Santiago de Chile. Av. Bernardo O'Higgins 3363, Santiago, Chile.
[2]University of Groningen, Wirdumerdijk 34, 8911 CE, Leeuwarden, The Netherlands.
[3]Center for Environmental Remote Sensing, Chiba University, 1-33 Yayoicho, Inage Ward, Chiba, 263-8522, Japan.
[4]Alfred-Wegener-Institute Helmholtz Center for Polar and Marine Research, Am Handelshafen 12, 27570 Bremerhaven, Germany
[5]University of Magallanes, Av. Manuel Bulnes 1855, 621-0427 Punta Arenas, Chile
[6]Western Washington University, 516 High St, Bellingham, WA 98225, USA.
[7]National Snow and Ice Data Center, Cooperative Institute for Research in Environmental Sciences, University of Colorado—Boulder, 449 UCB University of Colorado Boulder, CO 80309-0449 USA.
[8]Royal Netherlands Meteorological Institute (KNMI), Utrechtseweg 297, 3731 GA De Bilt, The Netherlands.
[9]Energy and Sustainability Research Institute Groningen (ESRIG), University of Groningen, Groningen, The Netherlands.
[10]Department of Earth System Science, Stanford University, 473 Via Ortega, Stanford, CA, 94305-2210, USA.

*Correspondence to*: Sarah Feron (s.c.feron@rug.nl)

**Abstract.** By influencing the circumpolar westerly winds, the stratospheric polar vortex has played a major role in the Antarctic surface climate in recent decades. However, the footprint of the polar vortex variability in the year-to-year changes of the Antarctic sea ice cover remains obscured. Here, we combine satellite retrievals and reanalysis data to study the response of the sea ice extent around Antarctica to changes in the polar vortex strength. We focused on the last two decades that have seen sharp changes in the stratospheric zonal flow, the tropospheric westerly winds, and the sea ice cover (the latter climbed to record highs in 2013 and 2014 before dropping to record lows in 2017 and 2022). We found that this unprecedented interannual variability has been noticeably influenced by the polar vortex dynamics. The signature of the stratosphere-troposphere coupling is apparent on recent all-time records (highs and lows) in the sea ice around Antarctica.

## 1 Introduction

Sea ice plays an important role in the World's energy balance by reflecting the incoming solar radiation back to space and by regulating heat and gas exchanges between the ocean and the atmosphere (Hobbs et al., 2016). As a crucial breeding habitat for marine biota, sea ice is also vital to coastal ecosystems in Antarctica (Wing et al., 2021).

Unlike the Arctic where the sea ice extent has shown a steady decline since the start of satellite observations (Lannuzel et al., 2020), sea ice extent surrounding Antarctica has not demonstrated a clear trend. Sea ice reconstructions from ice cores (Thomas et al., 2019; Dalaiden et al., 2021; Yang et al., 2021) and data assimilation (Fogt et al., 2021) suggest that the Antarctic sea ice

extent decreased throughout much of the early and middle twentieth century before rebounding in the late 1970s. Ocean
      warming and increased ice-shelf melt in Antarctic likely contributed to the period of sea-ice expansion that ended in 2014
      (Bintanja et al., 2013).

      The sea ice extent in Antarctic waters climbed during most of the satellite era (the post-1979 period) to record highs before
recently plummeting to record lows. Satellite observations detected record highs for winter sea ice extent for three consecutive
      years, from 2012 to 2014 (Meehl et al., 2016; Reid and Massom, 2015). However, considerable losses in sea ice around
      Antarctica started to occur in 2016 and led to record lows in 2017 (Parkinson, 2019; Turner et al., 2017) and in 2022 (Turner
      et al., 2022; Raphael and Handcock, 2022). The enhanced interannual variability that led to recent sea ice records (highs and
      lows) appears to be related to swift changes in the westerly winds that encircle Antarctica (Eayrs et al., 2021; Doddridge and
Marshall, 2017).

      A comparison with major climate indices shows that the long-term trends in the sea ice cover are largely dominated by the
      changes in the Southern Annular Mode (SAM) (Yang et al., 2021), which describes the strength and position of the circumpolar
      westerly winds (Mo, 2000). The positive (negative) SAM phase is associated with the strengthening (weakening) and poleward
(northward) migration of the westerly winds. While strengthened westerlies favor northward sea-ice transport and expansion
      (Eayrs et al., 2021; Ferreira et al., 2015), they also facilitate heat advection from the subtropics that may result in sea ice
      melting (Crosta et al., 2021). The trend toward positive values of the SAM index observed since the late 1950s (Marshall,
      2003) has played a central role in the recent Antarctic mass redistributions (Medley and Thomas 2019) and may have
      contributed to the expansion of the Antarctic sea ice cover observed from 1979 to 2015 (Yang et al., 2021).


      There is growing evidence suggesting that year-to year changes in the sea ice cover are determined by the variability of the
      circumpolar westerly winds (Eayrs et al., 2021; Doddridge et al., 2017). However, the processes that drive this variability are
      not fully understood. Through atmospheric teleconnections, changes in the South Pacific tropospheric circulation can be
      influenced by the positive (i.e., El Niño) and negative (i.e., La Niña) phases of El Niño-Southern Oscillation (ENSO) (Fogt et
al., 2011). Although they are not always concurrent, La Niña (El Niño) events coincide with positive (negative) phases of
      SAM more often than expected by chance (Fogt et al., 2011), which makes periods of strengthened westerlies more likely
      during La Niña (Eayrs et al., 2021).

      The tropospheric westerly winds are also affected by the strength of the stratospheric polar vortex. The downward impact of
weak (strong) polar vortex regimes has been detected in the subsequent shift of the SAM index to its negative (positive) phase
      (Bergner et al., 2022; Thompson et al., 2005). The connection between polar vortex strength and surface climate can be long
      lasting too; a strong stratospheric polar vortex is often followed by strengthened tropospheric westerlies (i.e., positive SAM
      values) that can persist for months (Lim et al., 2018).



By affecting the polar vortex, the stratospheric ozone depletion has played an important role in the Antarctic surface climate in recent decades. A cold (warm) and strengthened (disturbed) stratospheric polar vortex is associated with large (small) and long-lasting (short-lived) ozone holes (Cordero et al., 2022). The long-term trend toward positive values of the SAM index (Fig. S1a) has been partially attributed to the stratospheric ozone depletion that occurs over Antarctica every spring (Damiani et al., 2020; Polvani et al., 2011; Previdi and Polvani, 2014; Thompson et al., 2011). The pause (apparent in summertime only, Fig. S1b) in the long-term strengthening of the SAM has been credited to the success of the Montreal Protocol that banned human-made ozone-depleting substances (Banerjee et al., 2020).

Despite the well-established influence of the stratospheric polar vortex on the circumpolar westerly winds (i.e., the SAM phase), the footprint of the polar vortex variability in the year-to-year changes of the Antarctic sea ice cover remains elusive. Here, we combine existing satellite retrievals and reanalysis datasets to study the response of the sea ice extent around Antarctica to changes in the polar vortex strength. We focused on the last two decades that have seen sharp changes in the Antarctic polar vortex, the westerly winds, and the sea ice cover. We found that this unprecedented interannual variability has been noticeably influenced by the polar vortex dynamics. The signature of the stratosphere-troposphere coupling is apparent on recent all-time records (highs and lows) in the sea ice around Antarctica.

## 2 Data and Methods

Daily estimates of the sea ice extent and concentration data from November 1978 to the present were obtained from the U.S. National Snow and Ice Data Center Sea Ice Index (Fetterer et al., 2017). The sea ice extent corresponds to the area where the sea ice concentration equaled or exceeded 0.15. The data correspond to the platforms DMSP, DMSP 5D-3/F17, DMSP 5D-3/F18, and Nimbus-7, and the sensors SMMR, SSM/I, and SSMIS. Sea ice anomalies in sea ice concentration were calculated using a 30-year reference period from 1981 to 2010. Sea ice extent and concentration data are available at: https://nsidc.org/data/g02135/versions/3

Daily estimates of the SAM index were obtained from the Climate Prediction Center (National Weather Service, National Oceanic and Atmospheric Administration – NOAA). The daily SAM index is constructed by projecting the daily (00Z) 700mb height anomalies poleward of 20°S onto the loading pattern of the SAM (Mo, 2000). The loading pattern of the SAM is defined as the leading mode of Empirical Orthogonal Function (EOF) analysis of monthly mean 700 hPa height during 1979-2000 period. Anomalies in the SAM index were calculated using a 30-year reference period from 1981 to 2010. Daily estimates of the SAM index are available at: https://www.cpc.ncep.noaa.gov/products/precip/CWlink/daily_ao_index/aao/aao.shtml#publication



Daily estimates of the near-surface wind speed, near-surface temperature, the sea-level pressure, and the potential vorticity (100 hPa) come from the atmospheric reanalysis ERA5 produced by the European Centre for Medium-range Weather Forecasts (ECMWF) (Hersbach et al., 2020). Anomalies in the wind speed, the sea-level pressure, and the near-surface temperature were calculated using a 30-year reference period from 1981 to 2010. The potential vorticity is a conserved quantity that acts as a

tracer for motion on an isentropic surface. Here, the vortex edge was defined applying a $1.3 \times 10^{-5}$ K kg$^{-1}$ m$^2$ s$^{-1}$ threshold. Data of the near-surface wind speed, the near-surface temperature, the sea-level pressure, and the potential vorticity (100 hPa) are available at: https://www.ecmwf.int/en/forecasts/datasets/reanalysis-datasets/era5

Daily estimates of the 45-75° zonal mean zonal wind speed on the 100-hPa surface, the 60-90° zonal mean zonal temperature

on the 50-hPa surface, the ozone hole area, and the polar vortex area on the 460-K isentropic surface are from the Modern-Era Retrospective analysis for Research and Applications, Version 2 (MERRA-2) assimilation (Gelaro et al., 2017). The average east-west (zonal) wind speed for 45°S to 75°S is near the peak of the polar jet maximum. This jet stream isolates air over Antarctica from air in the midlatitudes. The region poleward of this jet stream is called the Antarctic polar vortex. The 60-90° zonal mean zonal temperature is the temperature averaged around the polar cap for latitudes south of 60°S. This is a good

measure of the overall temperature in the polar vortex. The ozone hole area is determined from total ozone satellite measurements and corresponds to the region of ozone values below 220 Dobson Units (DU) located south of 40°S. Values below 220 DU represent anthropogenic ozone losses over Antarctica. Anomalies in the 45-75° zonal mean zonal wind speed, the 60-90° zonal mean zonal temperature, and the polar vortex area were calculated using a 30-year reference period of 1981 through 2010. In the case of the ozone hole area, we excluded the last two decades of the last century, when accelerated growth

of the ozone hole area masked changes in the interannual variability. Accordingly, anomalies in the ozone hole area were calculated using a 30-year reference period of 2001 through 2010. Daily estimates of the 45-75° zonal mean zonal wind speed, the 60-90° zonal mean zonal temperature, the ozone hole area, and the polar vortex area are available at: https://gmao.gsfc.nasa.gov/reanalysis/MERRA-2/

## 3 Results

### 3.1 Recent Record

In February 2022, sea ice extent around Antarctica dropped to its record low since satellite monitoring began in 1978 (Fig. 1a). Sea ice retreat anomalies began in early September 2021 along the outer edges of the winter pack ice and proceeded southward (poleward) over the next 5 months. By late February 2022, the sea ice extent dramatically dropped to less than 2 million square kilometers, about 30% below the mean annual minimum (Turner et al., 2022). In addition to the all-time record

annual minimum on 25 February 2022, the Antarctic sea ice experienced its lowest annual mean extent since the start of satellite observations. This minimum surpassed the previous record low of 2017 (Fig. S2). Although annual minima may be



attributable to local atmospheric extremes operating on daily timescales (i.e., deep storms), low annual averages suggest the influence of persistent circulation anomalies associated with climate modes.

The most remarkable anomalies observed in 2022 included sea ice concentrations well below long-term averages in the Bellingshausen-Amundsen Sea sector, which was in sharp contrast to positive sea ice anomalies in the Ross Sea sector (Fig. 1b and Fig. S3). This regional pattern is compatible with a strengthened Amundsen Sea Low resulting from a high density of storms in the region southwest of the Antarctic Peninsula. The Amundsen Sea Low deepened in 2022 (Turner et al., 2022), which favored heat advection from the subtropics to the Antarctic Peninsula and the northward cold airflow from the Antarctic
Plateau to the Ross Sea sector. The strengthened Amundsen Sea Low seen in 2022 (Fig. S4) likely contributed to the sea ice negative (positive) anomalies in the Bellingshausen-Amundsen (Ross) Sea sector shown in Fig. 1b, and to the extremely warm year at the northern tip of Antarctica. The strength of the north to northwesterly airflow down to the Antarctic Peninsula contributed to 2022 being the warmest year on record in this region (Fig. S5).

The all-time records in the sea ice extent in 2022 were likely favored by persistently positive SAM values and by the return of La Niña. The positive phase of the SAM and La Niña strengthen the Amundsen Sea Low, which in turn affects the amount and meridional extent of sea ice in the Bellingshausen and Ross Sea sectors. While La Niña makes the Amundsen Sea Low stronger than average, a positive SAM leads to the poleward displacement of the cyclone tracks, which in turn also reinforces the Amundsen Sea Low (Clem et al., 2017). Both factors (La Niña and a positive phase of the SAM) were presented in 2022.
While La Niña returned in 2022 for the third year in a row, the mean annual SAM index ended in positive values for the third year in a row. In fact, the SAM index exhibited in 2022 the highest spring average since 2008 (Fig. S1c).

The strengthened Amundsen Sea Low, which played a major role in the 2022 records, was not a factor in 2017 (Fig. S4) when the Antarctic sea ice experienced its now second lowest annual mean extent. Instead of the tradeoff between the Bellingshausen-Amundsen and the Ross sectors seen in 2022, anomalous springtime retreat occurred in all sectors in 2017
(although larger in the Weddell and Ross Seas). Still, just like in 2022, the near-record sea ice losses in 2017 concurred with persistently positive SAM values (Fig. S1c).

## 3.2 Recent records seen in context

The record-breaking positive and negative anomalies in the sea ice during the last two decades are a sign of enhanced variability
around Antarctica (Fig. 1c). In a few years, the Antarctic sea ice extent climbed to record highs (from 2012 to 2014) before dropping to record lows (in 2017 and 2022). Taken as a measure of the variability, the standard deviation of the sea ice extent anomalies (Fig. 1c) climbed by about 75%, from 0.4 million square kilometers over the period 1979-2002 to 0.7 million square kilometers over the period 2003-2022. The enhanced variability that led to the recent sea ice records (highs and lows) appears to be related to swift changes in the westerly winds that encircle Antarctica (Eayrs et al., 2021; Doddridge et al., 2017).



The significant losses in the sea ice extent that started to occur around Antarctica in 2016 coincided with remarkably strong
tropospheric westerly winds in early spring. The strengthened westerly winds around Antarctica are the cause of the strong
positive values of the SAM index, which exhibited record values in early spring over the period 2016-2022 (Fig. 2a). Ranked
by the mean SAM index for September, five of the ten largest positive SAM anomalies on record occurred since 2016 (Fig.
S6). These SAM records may have been favored by the prevailing ENSO phase. Although they are not always concurrent,
positive phases of the SAM are more likely during La Niña (Fogt et al., 2011). The latter showed up 4 times over the period
2016-2022 (in 2018 and from 2020 to 2022).

Over the period 2016-2022, the average near-surface wind speed for September showed strong positive anomalies around
Antarctica (Fig. 2b), which were particularly large along the northern limit of the Bellingshausen Sea. These anomalies are
consistent with a strengthened Amundsen Sea Low. Although the Amundsen Sea Low has deepened since the mid-20th century
(Dalaiden et al., 2022), the recurrent La Niña events, along with persistently positive SAM values, likely contributed to
strengthening it further over the period 2016-2022. The strong positive wind speed anomalies for September 2016-2022 (Fig.
2b) are in sharp contrast with the slightly negative anomalies for September 2007-2015 (Fig. S7). Over the period 2007-2015,
the sea ice cover appeared to be expanding.


Time series of the SAM index and the sea ice extent during the last two decades suggest that the strength of the westerly winds
in early spring drives the subsequent sea ice retreat. Figure 2c shows that strong (weak) near-surface westerly winds in
September have often preceded negative (positive) anomalies in the sea ice extent in spring and early summer. The high
correlation between the SAM index in September and sea ice extent during the melt season (Fig. 2c) provides predictability of
sea ice cover, several months in advance. These results add evidence that there is a high degree of inter-seasonal persistence
in the sea ice retreat or advance (Stammerjohn et al., 2008) and that atmospheric circulation anomalies in spring can
precondition sea ice melt well ahead of the annual minimum (Holland et al., 2018).

While strengthened westerlies may favor springtime sea ice retreat, prior efforts have found that strong westerlies favor
wintertime sea-ice expansion (Ferreira et al., 2015). A positive phase of the SAM has been associated with the sea ice expansion
(Yang et al., 2021; Ferreira et al., 2015; Lefebvre and Goosse, 2008) that led to record highs for winter sea ice extent in 2013
and 2014 (Meehl et al., 2016; Reid and Massom, 2015). Although the mean SAM for June 2022 was the lowest since 2007, a
trend toward positive values is still apparent in the mean SAM index for early winter (Fig. S1f).

### 3.3 Stratosphere-troposphere coupling

The tropospheric westerly winds are influenced by the strength of the stratospheric polar vortex, which in turn is related to the
wind speeds in the polar-front jet stream (the polar vortex is the region poleward of the jet). Attributable to this stratosphere-
troposphere coupling, when the polar vortex is colder and stronger than usual, the near-surface westerly winds encircling



Antarctica strengthen and the SAM index becomes persistently positive. By contrast, a weak and relatively warm polar vortex is associated with weaker westerly winds around Antarctica and negative phases of the SAM index (Bergner et al., 2022; Thompson et al., 2005). As a result, there is a robust correlation (R>0.8) in winter and in spring between the SAM index and the 45-75° zonal mean zonal wind speed on the 100-hPa surface (Fig. 3a). The SAM index is also correlated in spring (R~-0.7) with the 60-90° zonal mean zonal temperature on the 50-hPa surface and with both the polar vortex (R~0.6) and the ozone hole area (R~0.7) (Fig. S8).

The tropospheric westerly winds can be also influenced by the effect of the ozone hole on the strength of the stratospheric polar vortex (Damiani et al., 2020; Polvani et al., 2011; Previdi and Polvani, 2014; Thompson et al., 2011). Although the long-term trend in the ozone hole area depends on the amounts of ozone-depleting substances in the atmosphere (WMO, 2018), the interannual variability of the ozone hole area is largely governed by the strength of the polar vortex. However, there is also a feedback loop between the strength of the polar vortex and the ozone hole. Since ozone is a greenhouse gas, its depletion has a cooling effect in the middle and upper stratosphere, which in turn reinforces the polar vortex and affects the tropospheric westerly winds. Roughly mirroring long-term changes in stratospheric temperature, the ozone hole area (the stratospheric temperature) increased (decreased) from the early 1980s until reaching its peak about two decades ago (Fig. S9). Although the trend is less clear in late spring, the ozone hole area (the stratospheric temperature) in early spring appears to be steadily declining (increasing) since the early 2000s due to the success of the Montreal Protocol (Cordero et al., 2022).

The stratosphere-troposphere coupling is apparent from the quasi-simultaneous changes in the tropospheric westerly winds and the stratospheric jet stream (Fig. 3b, lower panel shows the departures from the average tropospheric SAM index and the 45-75° zonal mean zonal wind speed on the 100-hPa surface). As shown in Fig. 3b (lower panel), the considerable Antarctic sea ice losses in recent years coincided with strong tropospheric westerly winds and a strengthened stratospheric jet stream in early spring (September). This suggests that the strong positive values of the SAM index around September over the period 2016-2022 were influenced by the strengthened jet, which in recent years exhibited relatively high values around September (Fig. S10). Ranked by the monthly mean for September, six of the top 15 highest mean 45-75° zonal mean zonal wind speeds since 1980 occurred during the last decade (Fig. S10). This strong polar vortex is likely one reason we have seen so many strongly positive SAM values for September in recent years (Fig. S6). The relatively strong stratospheric jet stream observed in early spring (September) since the mid 2010s is remarkable considering that they occurred during the decade when the Antarctic ozone hole healing emerged (Solomon et al., 2016). A strengthened stratospheric polar vortex is favored by low stratospheric temperatures but, in early spring, the stratosphere appears to be steadily warming due to the Antarctic ozone recovery (Fig. S9b).

It is unlikely that the ozone depletion provided a considerable contribution to the strong stratospheric jet stream seen in early spring (September) since the mid 2010s. Due to the Antarctic ozone healing, the significant losses in the Antarctic sea ice



extent in recent years coincided with relatively small ozone holes in early spring. As shown in Fig. 3b (upper panel), the ozone hole area in early spring over the period 2016-2022 was on average up to 3 million square kilometers smaller than the 2001-2010 mean. The small early spring ozone holes seen in recent years may have contributed to the concurrent positive anomalies

in the stratospheric temperature shown in Fig. 3b (middle panel). These positive anomalies point to relatively high stratospheric temperatures that do not favor a strengthened polar vortex. In other words, the ozone depletion seen in recent years in early spring cannot explain the relatively strong stratospheric jet stream observed since the mid 2010s. By contrast, the persistency of strengthened stratospheric winds did likely lead to the negative anomalies in the stratospheric temperature (up to 4°C below the 1981-2010 mean) and the relatively large ozone hole (about 1 million square kilometers larger than the 2001-2010 mean)

seen in late spring (November) in recent years (Fig. 3b, upper and middle panels).

### 3.4 Shapeshifting vortex's effects

The strong polar vortex observed in recent years (especially in early spring) likely played a role in the Antarctic sea ice losses that started to occur in 2016. As noted earlier, satellite observations demonstrated record lows in sea ice around Antarctica in 2017 (Parkinson, 2019) and in 2022 (Turner et al., 2022; Raphael and Handcock, 2022). Although these losses were of

comparable magnitude, they exhibited different regional patterns that may have been influenced by the shapeshifting polar vortex (Fig. 4a). On a regional scale, the most remarkable differences between the springtime sea ice anomalies seen in 2022 and 2017 occurred in the Ross Sea. Ice concentrations well above long-term averages in the Ross Sea sector in spring 2022 (Fig. 4a, upper panel) were in sharp contrast to anomalously negative sea ice anomalies in the same sector in spring 2017 (Fig. 4a, lower panel). Still, negative anomalies were detected in both the Weddell and Indian Seas in spring 2017 and spring 2022.

These regional patterns are compatible with anomalies in the Amundsen Sea Low (deeper in 2022 than in 2017; Fig. S4). Although the absolute depth of the Amundsen Sea Low is strongly influenced by the SAM phase (Raphael et al., 2016), differences between the SAM index in spring 2022 and in spring 2017 were not considerable (Fig. S6). Differences between wind speeds in the polar-front jet stream were not substantial either. What did exhibit considerable differences was the shape of stratospheric polar vortex (Fig. 4b).


The distinctive regional patterns associated with the sea ice anomalies shown in Fig. 4a are in general compatible with large-amplitude anomalies in the stratospheric zonal flow shown in Fig. 4b. Although roughly centered over Antarctica, the stratospheric polar vortex is often irregularly shaped with several regions where the cold air can shift northward and regions where the warm air can extend southward. Relative to 1981-2010 climatology, the early-spring vortex exhibited negative

anomalies in both the Weddell and Indian Seas (Fig. 4b). These negative anomalies allowed warm air to extend southward, likely contributing to the sea ice retreat in these regions in spring 2017 and 2022 (Fig. 4a). The early-spring vortex exhibited opposite anomalies in the Amundsen and Ross Seas (negative in 2017 and slightly positive in 2022; Fig. 4b). These anomalies allowed cold air to dip northward, likely contributing to the sea ice retreat in this region in spring 2017, as well as to the sea ice advance in the same region in spring 2022 (Fig. 4a).




The shapeshifting polar vortex in early winter (June) may have also played a role in the regional anomalies associated with the wintertime sea ice (Fig. S11). Although the sea ice extent reached record highs for winter in 2013 and 2014 (Meehl et al., 2016; Reid and Massom, 2015), modest negative anomalies were detected in the Weddell Sea sector in 2013 and along the northeastern Antarctic Peninsula in 2014 (Fig. S11). These small anomalies concurred with regions where the irregularly shaped polar vortex enable warm air to extend southward (Fig. S11). Coincidences between the polar vortex and the sea ice extent anomalies also occurred in the western Indian Ocean and the eastern Ross Sea where the polar vortex allowed cold air to shift northward, likely favoring the sea ice expansion in those sectors.

The coincidences noted above between the sea ice extent anomalies and large-amplitude anomalies in the stratospheric zonal flow suggest that the stratospheric circulation influences the regional configuration of sea ice advance and retreat. However, the available evidence is still far from being conclusive because on the one hand the record is limited, and on the other hand the year-to-year-anomalies detected since the start of satellite observations have not been that large. Despite being unprecedented, record highs for winter sea ice extent (in 2013 and 2014) and record lows for spring sea ice extent (in 2017 and 2022) were relatively modest (about +8% for winter records and about -7% for spring records). Modest anomalies are subjected to the influence of local/regional circulation anomalies, including individual storms in the melt season may be important in establishing negative sea ice records. Still, a discussion on the available evidence is worthy.

## 4 Discussion and Conclusions

The stratospheric polar vortex can lead to changes in the tropospheric circulation and impact the surface climate on a wide range of timescales. Here, we combine satellite retrievals and reanalysis datasets to study the role of the polar vortex variability in the year-to-year changes of the Antarctic sea ice cover. First, we address the connection between the stratospheric polar vortex dynamics and tropospheric westerly winds. Second, we explore the response of the sea ice extent around Antarctica to changes in the strength of the westerly winds. Our results suggest that changes in the often-coupled stratospheric zonal flow and the tropospheric westerly winds have likely contributed to the recent all-time records (highs and lows) in the Antarctic sea ice.


We found a strong correlation between the strength of the tropospheric westerlies and the wind speeds in the polar-front jet stream (the polar vortex is the region poleward of the jet), which denote a robust stratosphere-troposphere coupling. The quasi-simultaneous stratospheric and tropospheric anomalies seen in recent years over Antarctica underline the influence of the stratospheric circulation on the interannual variability of the Antarctic surface climate. These results add to a growing body of evidence that suggests that stratospheric variability plays an important role in driving the near-surface climate (Lim et al., 2018; Byrne and Shepherd, 2018).



Due to the stratosphere-troposphere coupling, the strong stratospheric jet stream frequently seen around September in recent years has likely contributed to strengthening the tropospheric westerly winds around Antarctica. Changes in the westerlies

have huge impacts on the Antarctic surface climate as the strength of the westerly winds drives the baffling rise and fall of sea ice cover around Antarctica. While prior efforts have shown that strong westerlies favor wintertime sea-ice expansion (Ferreira et al., 2015), our results suggest that strengthened westerlies may favor springtime sea ice retreat. The significant losses in the sea ice extent that started to occur around Antarctica in 2016 coincided with remarkably strong tropospheric westerly winds in early spring.


We found that strong (weak) near-surface westerly winds in September have often preceded negative (positive) anomalies in the sea ice extent in spring and early summer. The high correlation between the strength of the westerly winds in September and sea ice extent during the melt season provides predictability of sea ice cover several months in advance. These results add to evidence that there is a high degree of interseasonal persistence in the sea ice retreat or advance (Stammerjohn et al., 2008)

and that atmospheric circulation anomalies in spring can precondition the sea ice melt well ahead of the annual minimum (Holland et al., 2018).

We also found the footprints of the stratospheric circulation (i.e., the shapeshifting polar vortex) in the distinctive regional patterns associated with the recent all-time record lows in the sea ice around Antarctica. The stratospheric polar vortex often

exhibits regions where the cold air can shift northward (contributing to sea ice advance) and regions where the warm air can extend southward (contributing to regional sea ice retreat). Although we found that large-amplitude anomalies in the stratospheric zonal flow generally compatible with the regional configuration of sea ice advance and retreat, the influence of local/regional circulation anomalies, including individual storms, cannot be ruled out. Thus, as noted above, assessing the robustness of this connection requires a longer record.


Our results suggest that, by influencing the circumpolar tropospheric westerly winds, a strengthened polar vortex in early spring may have played a role in the recent all-time record lows in the sea ice around Antarctica. So, the question arises: what has caused the polar vortex to strengthen? Addressing this question will require further cross-disciplinary work and international collaboration. However, our analysis provides some clues. Although it seems probable that more than one factor

was responsible, the ozone hole is unlikely to be one of them. The cooling signal associated with ozone depletion reinforced the vortex until the early 2000s, but this effect started to reverse due to the onset of the ozone healing. Attributable the ongoing ozone recovery, the ozone hole is not likely contributing to the strong stratospheric jet stream seen in early spring (September) since the mid 2010s. After ruling out the ozone depletion, the main drivers behind the early spring strengthening of the polar vortex seen in recent years maybe in the tropics. A number of studies have examined the year-to-year sea ice variability of the

sea ice cover and suggested it is linked to a range tropical forcing factors (Holland et al., 2017; Meehl et al., 2016; Stammerjohn et al., 2008).

ENSO influences the tropical atmospheric circulation leading to large-scale planetary waves that affect the strength of the polar vortex (Kwon et al., 2020; WMO, 2018). However, our understanding of the processes that determine the interannual

variability of the planetary wave activity is still incomplete and the effects of climate change on ENSO are not clear. What is clear is that recent changes in the Antarctic sea ice has been driven by a near-surface atmospheric circulation influenced by large-amplitude anomalies in the stratospheric zonal flow.

**Data Availability**

Sea ice extent and concentration data are available at: https://nsidc.org/data/g02135/versions/3


Daily estimates of the SAM index are available at: https://www.cpc.ncep.noaa.gov/products/precip/CWlink/daily_ao_index/aao/aao.shtml#publication

Data of the near-surface wind speed, the near-surface temperature, the sea-level pressure, and the potential vorticity are

available at: https://www.ecmwf.int/en/forecasts/datasets/reanalysis-datasets/era5

Daily estimates of the 45-75° zonal mean zonal wind speed, the 60-90° zonal mean zonal temperature, the ozone hole area, and the polar vortex area are available at: https://gmao.gsfc.nasa.gov/reanalysis/MERRA-2/

**Author contributions**

Conceived and designed the experiments, R.R.C., and S.F.; analyzed the data, R.R.C., A.D., P.J.L., J.C., and S.F.; contributed materials/analysis tools, A.L.K., and S.F.; writing—review and editing, R.R.C., S.F., A.D., R.B., Z.O., and G.C.; all authors have read and agreed to the published version of the manuscript.

**Competing interests**

The authors declare that they have no competing interests.



**Acknowledgements**

We thank the Laboratory for Atmospheres at NASA's Goddard Space Flight Center, the European Centre for Medium-range Weather Forecasts (ECMWF), as well as each of the satellite teams for the data access and all their hard work in producing the datasets. We also thank the researchers contributing to the National Snow and Ice Data Center Sea Ice Index.

**Financial support**

The support of ANID (ANILLO ACT210046 and FONDECYT 1231904) and INACH (RT_69-20) is gratefully acknowledged.

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





**Figure 1: The sea ice extent in Antarctic waters increased during most of the satellite era to record highs before recently dropping to record lows.**

**(a) Daily sea ice extent during the season 2021-2022 (red line). The gray shading indicates the highest and lowest daily values measured over the period 1981–2010 while the white line indicates the mean over the same period. The sea ice extent corresponds to the area where the sea ice concentration equaled or exceeded 0.15. Antarctica is surrounded by seasonally varying sea ice extent, which typically reaches about 18.5 million square kilometers at the maximum in late September, shrinking to around 2.5 million square kilometers in late February or early March. The Antarctic sea ice dropped to its all-time record low in February 2022.**

**(b) Sea ice concentration for 2022 and mean ice edge averaged over the period 1981–2010 (red line). The ice edge was defined by applying a 0.15 threshold. The Antarctic sea ice extent in 2022 was the lowest annual mean since the start of satellite observations. Relative to the 1981-2010 climatology, sea ice negative (positive) anomalies are apparent in the Bellingshausen (Ross) Sea sector.**

**(c) Daily sea ice extent relative to the 1981–2010 mean. The bold red line shows the 4-year centered moving average. Satellite observations indicate record highs for winter sea ice extent from 2012 to 2014. Record highs were followed by losses starting in 2016,**
**and record annual minima in 2017 and 2022. The daily sea ice extent anomalies exhibited an enhanced variability during the last two decades.**

**Sea ice data were obtained from the U.S. National Snow and Ice Data Center Sea Ice Index (Fetterer et al., 2017). Plots were generated using Python's Matplotlib library (Hunter, 2007).**




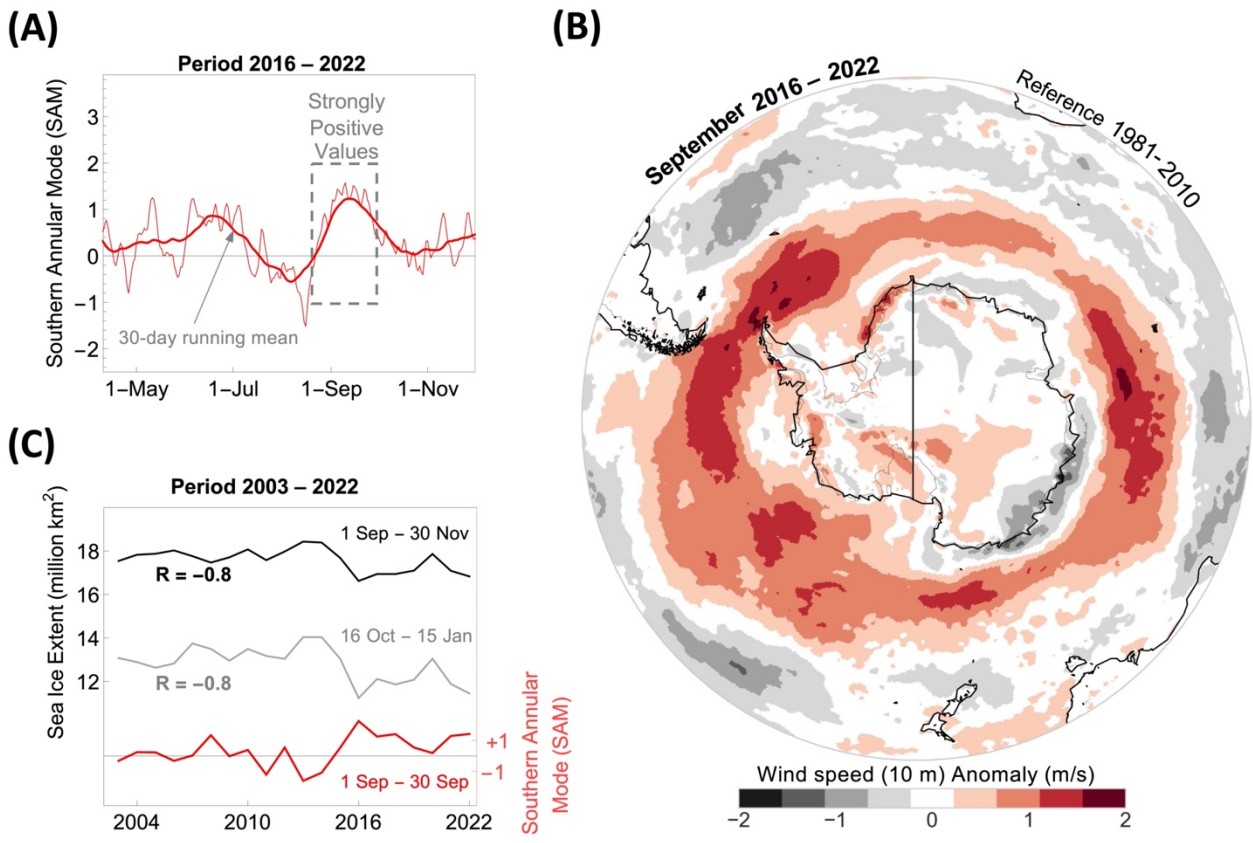

**Figure 2. The strength of the westerly winds in early spring plays a major role in the sea ice retreat during the melt season.**

**(a) Daily estimates of the Southern Annular Mode (SAM) index averaged over the period 2016-2022 (thin red line). The bold red line shows the 30-day centered moving mean. Over the period 2016-2022, the SAM index exhibited strongly positive values in early spring (September). Ranked by the mean SAM index for September (early spring), five of the ten largest positive SAM anomalies occurred over the period 2016-2022.**

**(b) Average near-surface wind speed for September 2016-2022, relative to the 1981–2010 mean. The strengthened westerly winds around Antarctica in early spring (September) over the period 2016-2022 are consistent with the strongly positive values of the SAM index shown in (a).**

**(c) Progress of the SAM index averaged from 1 September to 30 September (red line) and the sea ice extent averaged from 1 September to 30 November (black line) and from 16 October to 15 January (gray line). The high correlation (R=-0.8) between the SAM index in September and the sea ice extent during the melt season, suggests that the strength of the westerly winds in early spring can precondition the sea ice retreat several months in advance.**

**Daily estimates of the SAM index were obtained from Climate Prediction Center (National Weather Service, National Oceanic and Atmospheric Administration – NOAA (Mo, 2000)). Data of the near-surface wind speed comes from the atmospheric reanalysis ERA5 produced by the European Centre for Medium-range Weather Forecasts (ECMWF) (Hersbach, 2016). Sea ice data were obtained from the U.S. National Snow and Ice Data Center Sea Ice Index (Fetterer et al., 2017). Plots were generated using Python's Matplotlib library (Hunter, 2007).**



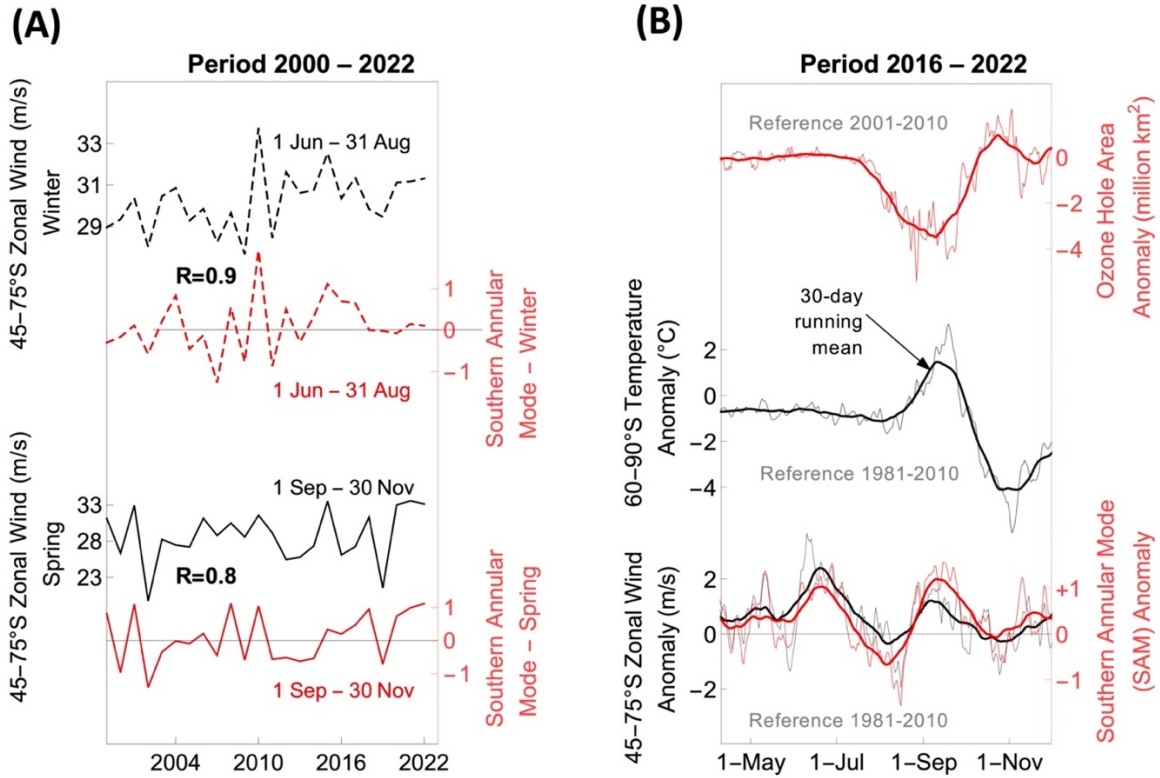

**Figure 3. Quasi-simultaneous stratospheric and tropospheric anomalies denote a robust stratosphere-troposphere coupling.**

555    **(a) Spring averages of the SAM index (red line) and of the 45-75° zonal mean zonal wind speed on the 100-hPa surface (black line). Winter averages of the SAM index (red dashed line) and the 45-75° zonal mean zonal wind speed on the 100-hPa surface (black dashed line). The average east-west (zonal) wind speed for 45°S to 75°S is near the peak of the polar jet maximum. The correlation coefficients (R) between the SAM index and the stratospheric wind speed are also shown in the plot.**

   **(b) Departures from the average Southern Annular Mode (SAM) index (red line, lower panel), 45-75° zonal mean zonal wind speed**
560    **on the 100-hPa surface (black line, lower panel), 60-90° zonal mean zonal temperature on the 50-hPa surface (black line, middle panel), and ozone hole area (red line, upper panel). Thin lines show daily anomalies averaged over the period 2016-2022, while bold lines show the 30-day centered moving mean. Anomalies are relative to the 1981–2010 mean except for the ozone hole area anomalies that are relative to the 2001–2010 mean.**

   **Daily estimates of the SAM index were obtained from Climate Prediction Center (National Weather Service, National Oceanic and**
565    **Atmospheric Administration – NOAA (Mo, 2000)). The data for the 45-75° zonal mean zonal wind speed, the 60-90° zonal mean zonal temperature, and the ozone hole area, are from the Modern-Era Retrospective analysis for Research and Applications, Version 2 (MERRA-2) assimilation (Gelaro et al., 2017). Plots were generated using Python's Matplotlib library (Hunter, 2007).**





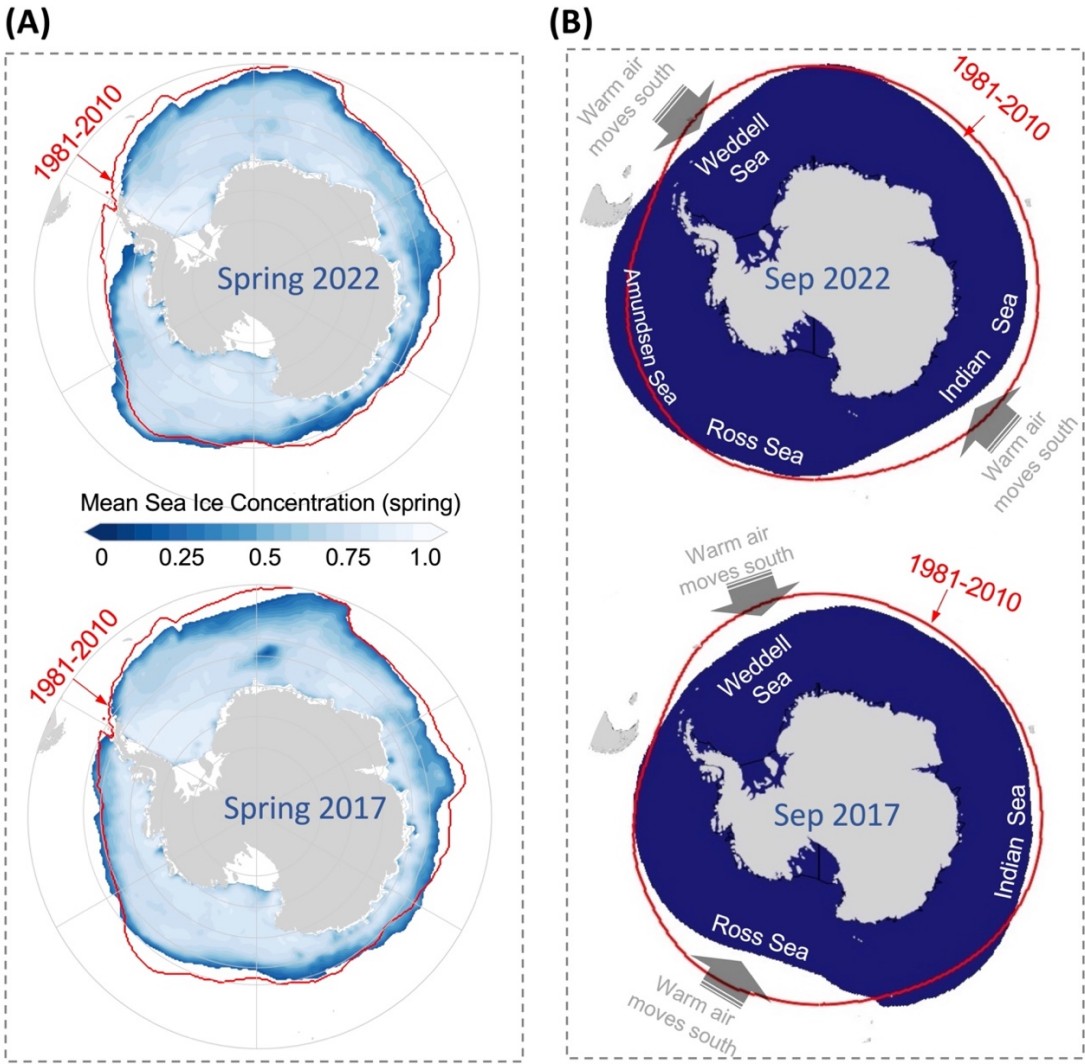

**Figure 4. The distinctive regional patterns associated with the springtime sea ice anomalies are in general compatible with large-amplitude anomalies in the stratospheric zonal flow.**

(a) Sea ice concentration for spring 2022 (upper panel) and for spring 2017 (lower panel). The spring ice edge averaged over the period 1981–2010 is also shown (red line). The ice edge was defined applying a 0.15 threshold. Relative to 1981-2010 climatology, there are several anomalies apparent. Spring 2022: positive anomalies in the Ross Sea and negative anomalies in the Weddell, Indian and Bellingshausen Seas. Spring 2017: negative anomalies in the Weddell, Indian and Ross Seas.

(b) Polar vortex (100 hPa) for September 2022 (upper panel) and for September 2017 (lower panel). The September vortex edge averaged over the period 1981–2010 is also shown (red line). The vortex edge was defined applying a 1.3 x $10^{-5}$ K kg$^{-1}$ m$^2$ s$^{-1}$ threshold to the potential vorticity on the 100 hPa pressure surface. Relative to 1981-2010 climatology, there are several anomalies apparent. Early spring 2022: positive anomalies in the Ross and Amundsen Seas and negative anomalies in the Weddell and Indian Seas. Early spring 2017: negative anomalies in the Weddell, Indian and Ross Seas.

Sea ice data were obtained from the U.S. National Snow and Ice Data Center Sea Ice Index (Fetterer et al., 2017[)] while potential vorticity data comes from the atmospheric reanalysis ERA5 produced by the European Centre for Medium-range Weather Forecasts (ECMWF) (Hersbach, 2016). Plots were generated using Python's Matplotlib library (Hunter, 2007).