# Peer review of "Signature of the stratosphere-troposphere coupling on recent recordbreaking Antarctic sea ice anomalies"

_The Cryosphere, 2023_

## Author Comment (AC1)

**Comments of Anonymous Referee #1**

**Comment of Reviewer 1**
The manuscript presents a link between the stratospheric polar vortex and recent record low Antarctic sea ice anomalies courtesy of a strong coupling between the tropospheric westerlies and the polar-front jet stream. Coincident wind and sea ice anomalies are shown, and some interesting insights emerge from the close inspection of variables considered (though sadly, many of these are in the Supplementary Material rather than the manuscript proper). As the changes to recent Antarctic sea ice behaviour continue to be a topic of high interest, the main concept of this manuscript is a thought-provoking and valuable angle of exploration. The analysis starts strong, but appears to lose confidence when attempting to correlate the shape of the vortex to sea ice anomalies, with the conclusions managing to be both tentative and brash at times. The findings are interesting, but not yet robust enough. Furthermore, the manuscript is sparsely sourced in many places, putting the onus on the reader to research much of the context and background. It is unclear why so much material is in the Supplementary Material compared to the 4 Figures in the manuscript proper, but the text of the manuscript could easily be reduced substantially to make more room for some of this additional material. Several parts of the results and discussion sections felt a bit disjointed and awkward and could be shortened. The general structure is adequate, but much of the manuscript would benefit from consideration as to whether the flow and phrasing is optimal, a check for missing words and duplication, as well as a sweep for clarity and conciseness. There is considerable potential for this work, and I therefore recommend it for reconsideration after major revisions, with the hope that the authors will find my following comments useful.

We thank the reviewer for her/his helpful comments and constructive remarks. We agree with the reviewer that interesting insights have emerged from the investigation that we carried out on the quasi-simultaneous evolution of sea ice cover, the tropospheric westerlies, and the polar-front jet stream. As pointed out by the reviewer, the main concept of our manuscript is a thought-provoking and valuable angle of exploration.

The reviewer argues that our conclusions are "both tentative and brash at times". It is an interesting point, and we thank the reviewer for raising it. Certainly, there are some aspects of the study that are more conclusive since they are built upon previously published results. For example, there is prior convincing evidence regarding the stratosphere-troposphere coupling. However, there are other aspects of our study that we have to be cautious about, since we are challenging a commonly accepted hypothesis. For example, the dynamics of the polar vortex is often assumed to not be consequential for Antarctic sea ice extent. In our manuscript we challenge this hypothesis with evidence that strongly suggests that the stratospheric polar vortex dynamics do play a role in the year-to-year changes of the Antarctic sea ice cover. Although our manuscript will hardly close the case, we do expect that it will trigger the cross-disciplinary work required for further exploration on the effects of the polar vortex dynamics on Antarctic sea ice extent. In the revised version of our paper, we will do our best to appropriately scale the emphasis of our conclusions.

We also agree with the reviewer on the significance of some of the information currently shown as "Supplementary Material". We are happy to incorporate his/her suggestions in this regard. As further discussed below, we believe that the clarity and conciseness of our manuscript will be greatly improved by the reviewers' remarks and suggestions.

**General Suggestions of Reviewer 1**
Check that the tenses of consecutive sentences is consistent (e.g. line 80-85). Section 2 (Data and Methods) is too long and wordy. It could easily be condensed down by stating at the top that 'daily estimates' of all data are used, and the reference period from which mean and anomalies are calculated

(except in the case of ozone), and then just stating where each dataset is from, with careful consideration of the most efficient presentation of any remaining information.

Figure captions tended to be too long and with too much detail. They could easily be shortened and tightened up. The submission policies for TC state that, as long as any abbreviations (e.g. datasets used) are already defined in the text, they do not need to be explicitly re-defined in the caption. After a concise description of the actual plot, any extra information would be better placed in the introduction and data/method sections – and I don't think necessary to include the programming package used to create each Figure at the end of each Figure. Perhaps an acknowledgement is a better option.

According to the suggestions of the reviewer, in the revised version of our manuscript, we propose to

- Check the consistency of the tenses of consecutive sentences (e.g., line 80-85).

- Shorten Section 2 (Data and Methods) stating at the top that 'daily estimates' of all data are used, and the reference period from which the mean and anomalies are calculated (except in the case of ozone). As suggested by the reviewer, in the new Section 2 we will just state where each dataset is from, carefully considering the most efficient presentation of the information.

- Shorten and tighten up Figure captions using abbreviations when possible (e.g., datasets used) and reducing the degree of details. After a concise description of the actual plot, any extra information will be placed in the introduction and data/method sections. As suggested by the reviewer, the programming package used to create the Figures will be included in the Acknowledgements.

**Line-by-Line Suggestions of Reviewer 1**
**Line 87:** It is common practice for sea ice extent to be defined in the data/method section, and thereafter referred to by the abbreviation of SIE in the following sections for efficiency.
**Line 101:** Can any of these variables be abbreviated or denoted using a symbol to be more efficient, since they're used multiple times in this (and other) paragraphs?
**Line 109:** There are almost no references in this section, other than for the data itself. Are these domains from the literature, or defined by the authors? The text states that average east-west wind speed between 45°S-75° is near the peak of the polar jet maximum, but the peak itself could be defined here too (and referenced for readers wanting to know more). The text states 'this is a good measure of the overall temperature of the polar vortex' – how do we know? How much variation occurs in zonal mean zonal temperature across the 60-90° domain? Is there a specific reference here for non-specialists?
**Line 128-131:** There is some redundancy here in highlighting the record minimum – a single (or two shorter) sentence/s could be crafted to incorporate this information more efficiently.
**Line 131:** Attributed by whom? Is there a study to which you are referring here? And why do low annual averages suggest the influence of persistent circulation anomalies of climate modes? Aren't the influences of some of these modes seasonal too?
**Line 136:** The text discusses sector-based domains, but the map only names four locations. If common sectors (ABS, RAS etc) are discussed, it would be helpful to clearly define these sectors both in the text and with lines on the map, clearly stating the place names for all sectors and sub-sectors of note.
**Line 145:** The SAM and La Nina discussion comes out of nowhere here. They are briefly discussed in the introduction, but no references are given explicitly here. In fact, between lines 145-157 could be reorganised and reworded entirely for clarity and flow. I really like Figure S4, and wonder if more could be made of it - perhaps incorporated into the body of the manuscript rather than supplementary material. I note the ASL is discussed more at lines 174-179 and again at 250-252; perhaps this would be a good place to incorporate an additional Figure to robustly illustrate your point.
**Line 159:** This states that the enhanced temporal variability occurs 'around Antarctica' – but is this also reflected in spatial variability?

**Line 156** states that the 2017 record event was due to a circumpolar pattern of ice loss, whereas the 2022 record was due to regionally opposing anomalies. Is spatial variability also enhancing over this period, or staying roughly the same?

**Line 164:** Firstly, this line is a direct repeat of lines 43-45. Also, the reference is Doddridge & Marshall, 2017 not Doddridge et al. 2017. Furthermore, the aforementioned reference states that the effect of wind anomalies on sea ice in 2017 was due to the negative SAM in 2016/17 summer, but so far in the manuscript the only discussion has been of persistently positive SAM anomalies. Figure S6 only shows September SAM index, where this negative polarity does not appear. Also, both these references discuss the effect of wind anomalies on the ocean and the combined effect on sea ice, whereas I do not see discussion of these important coupled processes in this manuscript.

**Line 166:** Are the strengthened westerly winds really the cause of the positive SAM?

**Line 168:** Does this hold true only for September? Is there a similar ranking of SAM anomalies for August, or October?

**Line 189-190:** Reconsider routine use of the word 'favor' throughout the manuscript. Reworking this paragraph to include a brief, simple description of why the same driver has opposing seasonal effects would be more impactful.

**Line 222-224:** I'm not enamoured by the argument that, of 42 years, 6 of the highest 15 occurred in the past 10 years. Why choose the top 15 highest? A qualitative look at Figure S10b tells me that 3 of those years were in the 80s, 5 were in the 90s, a single year was in the 2000s, and 6 occurred post-2015. If you only chose the top 10, the 2000s and beyond would barely feature compared to the 90s. I think there is probably a better way to present your argument, as this feels less than robust.

**Line 224-231:** I'm having trouble following the flow here. The strength of the jet since the mid-2010s is remarkable because of the healing ozone hole, but it is unlikely that ozone depletion provided a considerable contribution to it? In fact, the entire paragraph 230-240 is difficult for me to follow. Consider rewording for clarity and flow.

**Line 256:** In general, yes, compatible. But there are notable differences, which are not discussed. For the 2022 case, the Bellingshausen and Amundsen seas region are a stark contrast, and even the Ross Sea region to a lesser extent is not closely matched. In the 2017 case, the anomalies match more closely. Is it possible to condense these plots down? Does (b) really need shading? Perhaps the mean vortex line could be overlaid in a different colour to the mean extent line on the ice concentration plots, and stippling instead of shading for the vortex pattern? It would make it much easier to compare them rather than shifting constantly between panels.

**Line 279-281:** This feels overly pithy, and would benefit from rewording for clarification and flow. It gives a sort of 'shrug' impression to the end of the results section, which is oddly deprecating.

**Line 291:** I've seen this 'all-time records' a couple of times in the manuscript, and I think it would be better to frame this as according to the satellite data record, which only spans 43 years.

**Line 301-302:** Why are these underlined? It's distracting. If worried that the reader will not pick up on the difference between the two words, perhaps word it differently so it is clearer.

**Line 303:** I'm not sure about this. Technically they aren't remarkably nor consistently strong compared to previous decades, at least according to Figure S10 – the value for 2021 is the highest but 1982 was almost as high, and 2016, 2018 and 2020 rank well under the 5 ranking years from the 90s. 2017 doesn't even make the table. Compared to the 2000s, yes, the values are remarkably stronger, but is that what is meant here?

**Lines 321-331:** Requires editing and reworking – several adjoining words are missing and others are superfluous. The flow is difficult, and the conclusion seems hesitant. By contrast, the conclusion of lines 335-7 seems overly confident based on the evidence presented, particularly with no discussion of the effects of SST nor much of a mention of spatially heterogeneous or competing drivers.

According to the suggestions of the reviewer, in the revised version of our manuscript, we propose to

- Line 87: Define the sea ice extent in the data/method section and use thereafter the abbreviation of SIE.
- Line 101: Use abbreviations for improving efficiency.

- Line 109: Clarify that using these domains is a common practice and provide references supporting our statements regarding the peak of the polar jet maximum and the overall temperature of the polar vortex.
- Line 128-131: Reword these lines avoiding redundancies.
- Line 131: Cite prior studies that have studied first the effect of deep storms on annual SIE minima, and second the influence of climate modes (operating on seasonal or annual timescales) on the annual SIE mean.
- Line 136: Identify in the Figures common sectors (ABS, RAS etc) discussed in the text.
- Line 145: Reorganize and reword lines 145-157 for clarity and flow and provide references supporting our discussion involving SAM and La Nina.
- Incorporate Figure S4 into the body of the manuscript. As noted by the reviewer, this Figure can support the point we make in lines 174-179 and 250-252.
- Line 159: Clarify that the text in this line refers to the temporal variability only.
- Line 156: Clarify that, although the 2017 and 2022 SIE anomalies concurred with persistently positive SAM values, they exhibited considerably different regional patterns.
- Line 164: Avoid needless repetitions and correct reference (from Doddridge et al. 2017 to Doddridge & Marshall, 2017).
- Clarify that while Doddridge & Marshall (2017) studied the effects of SAM anomalies in the austral summer, we focused on the effects of SAM anomalies in the austral spring. We agree with the reviewer that both references in line 164 (Eayrs et al., 2021; Doddridge & Marshall, 2017) address points beyond the scope of our paper. However, both references underline the influence of the SAM and the westerly winds on the SIE, which is one of the main points made in our manuscript.
- Line 166: Rephrase the text avoiding any suggestion of causality.
- Line 168: Confirm that similar records have been seen in spring months (Fig. S1c) but not in winter months (Fig. S1e).
- Line 189-190: Avoid the use of the word 'favor' and rework this paragraph to include a brief description of why the same driver has opposing seasonal effects would be more impactful.
- Line 222-224: Eliminate the unnecessary argument regarding the strength of the jet. We agree with the reviewer that it is not particularly compelling.
- Line 224-231: Rephrase the paragraph 230-240 for clarity. According to the comment of the reviewer, we will clarify that although the ozone depletion provided a considerable contribution to strengthening the jet stream in the past, the ozone healing observed since the mid-2010s has likely erased that contribution.
- Line 256: Discuss the notable differences shown in Fig. 4 between the regional patterns associated with the springtime sea ice anomalies and large-amplitude anomalies in the stratospheric zonal flow.
- Explore different variants for Fig. 4 following the helpful suggestions of the reviewer. We will try to make it easier to compare plots a) and b).
- Line 279-281: Reword these lines for clarification. We appreciate the comment of the reviewer. It is of course not our intention to be self-deprecating, but we believe that it is important to be transparent regarding the limitations of the available evidence.
- Line 291: Avoid the repetitive use of the expression 'all-time records'. According to the suggestions of the reviewer, we will frame this as according to the satellite data record, which only spans 43 years.
- Line 301-302: Eliminate underlined text and make the text clearer.
- Line 303: Link the text in line 303 with Fig. 2a that shows that the significant losses in the sea ice extent that started to occur around Antarctica in 2016 coincided with remarkably strong tropospheric westerly winds in September.
- Lines 321-331: Edit these lines paying special attention to the adjoining words.
- Better calibrate the emphasis of our conclusions and explicitly state some of the limitations of this work, such as the effects of spatially heterogeneity or competing drivers, like the SST).

**Suggestion of Reviewer 1 (Figures)**

**Figure 1a:** I found it difficult to make sense of the arrows here. Why is the minimum arrow pointing near to the sea ice maximum, while the maximum arrow pointing towards the summer? I think the caption is a better place for this detail, as the presence and positioning of the arrows detracts from its readability.
In addition, I wonder if higher density is warranted on the X axis, and perhaps gridlines to show the start of each month rather than just a rough indication of each 3-month period. The text discusses late February and early September, but only shows a little marker for 1-Jan and 1-Oct. I'm not a big fan of the stretched Y axis either; I guess it shows the minimum more clearly but loses a lot of detail in the early September period which is also discussed in the text.

**Figure 3b:** Again, why not remove the arrow and simply state the 30-day running mean beneath the reference period in the legend for this part of the Figure? It is already stated in the caption that the bold line is the 30-day centered moving mean.

**Figure S4:** This seems an important and clear difference between the two years, though wind vectors overlaid on both plots to show anomalous wind directions resulting from the pressure anomalies would be a welcome addition.

According to the suggestions of the reviewer, in the revised version of our manuscript, we propose to

- Eliminate arrows in Fig. 1a and use the caption for details.
- Include gridlines to show the start of each month in Fig 1a.
- Explore different variants for the Y axis of Fig. 1a.
- Eliminate arrows in Fig. 3b and use the caption for details.
- Include wind vectors in Fig. S4.

We thank the reviewer once again for their helpful suggestions and constructive remarks. We believe these revisions will greatly improve the clarity and conciseness of our manuscript. As explained above, the revised version of our manuscript will be modified accordingly.

Best regards,

The authors.

---

## Author Comment (AC2)

**Final author comments (ACs); manuscript tc-2023-59**

Comments of Anonymous Referee #2

> **General Suggestions of Reviewer 2**
> This manuscript tackles the perplexing behavior of Antarctic sea ice for 2013-2022, especially the remarkable retreat during 2016-2022. Following the slow but steady expansion of sea ice extent since the start of the satellite record in 1979, a period of greatly enhanced variability commenced in 2013. Thus, there is high scientific merit in seeking to understand this unexpected behavior. That being said, I found this manuscript challenging to understand even after reading it twice. The broad range of material tackled contributes to my understanding challenge, likely due to my unfamiliarity with some aspects of sea ice behavior. Also in a very unusual departure, the analysis draws substantially on the supplementary material; the authors should consider moving the most important figures into the main manuscript. Mostly I would classify the specific comments below as a major-minor revision.

We thank the reviewer for her/his helpful comments and constructive remarks. We agree with the reviewer that there is high scientific merit in seeking to understand the recent record-breaking Antarctic sea ice anomalies.

We completely agree with the reviewer that the manuscript covers a "broad range of material". Certainly, studying the response of the sea ice extent around Antarctica to changes in the polar vortex strength is complex, which can make some parts of the text challenging for certain readers. However, we believe that the reviewers' remarks and suggestions will allow us to greatly improve the flow and phrasing our manuscript making the reading experience informative and rewarding.

We also agree with the reviewer on the significance of some of the information currently shown as "Supplementary Material". We are happy to incorporate all his/her suggestions in this regard.

> **Specific Suggestions of Reviewer 2**
> Page 4: Why did you use ERA5 for some variables and MERRA2 for others?
> Line 122: 10 year (2001-2010) reference period for ozone hole area?
> Line 166: "associated with" rather than "the cause of"?
> Line 263: How can "cold air to dip northward" lead to "sea ice retreat" in spring 2017? Similarly, the explanation for the sea ice advance in spring 2022 is unconvincing. These comments refer to the Ross Sea and Amundsen Sea region.
> The "shapeshifting vortex's effects" (Section 3.4) rely on the stratospheric anomalies to infer the surface temperature advection. You should show the actual surface circulation anomalies from ERA5 to back up your argument relating low-level air temperatures to sea ice anomalies.
> Lines 270-272: I don't see this association.
> Lines 327-330: You are left with unknown cause(s) of the "early spring strengthening of the polar vortex since the mid 2010s" that you have linked with the retreat of the sea ice cover 2016-2022.
> Figure 2b: Are these anomalies statistically significant?

According to the suggestions of the reviewer, we will revise our manuscript in the following ways:

- Line 87: Explain why we used both ERA5 and MERRA2.
- Line 122: Correct the typo in Line 122.
- Line 166: Rephrase line 166 according to the suggestion of the reviewer.
- Line 263: Rephrase the text in order to clarify that the comments refer to the Ross Sea and Amundsen Sea region. As shown in Fig. 4, anomalies in this region allowed cold air to dip

northward in Sep 2022, likely contributing to the sea ice advance in the Amundsen region in spring 2022.
- Add a new figure (based on reanalysis data) relating low-level air temperatures and circulation anomalies. We expect that this new figure will allow us to further analyze the effects of the shapeshifting vortex on the air surface temperature and in turn on the sea ice cover.
- Lines 270-272: Rephrase the unclear association described in text Lines 270-272.
- Lines 327-330: Expand upon the limitation of our work and the need for further research on the influence of other competing drivers.
- Figure 2b: Provide additional statistical information for assessing the significance of the relationships shown in Fig. 2b.

We thank the reviewer once again for her/his helpful suggestions and constructive remarks that we believe will greatly improve the clarity of our manuscript. As explained above, the revised version of our manuscript will be modified to address all the points that have been raised.

Best regards,

The authors.

---

## Author Response (AR1)

**Comments of Anonymous Referee #1**
* * *
**Comment of Reviewer 1**

The manuscript presents a link between the stratospheric polar vortex and recent record low Antarctic sea ice anomalies courtesy of a strong coupling between the tropospheric westerlies and the polar-front jet stream. Coincident wind and sea ice anomalies are shown, and some interesting insights emerge from the close inspection of variables considered (though sadly, many of these are in the Supplementary Material rather than the manuscript proper). As the changes to recent Antarctic sea ice behaviour continue to be a topic of high interest, the main concept of this manuscript is a thought-provoking and valuable angle of exploration. The analysis starts strong, but appears to lose confidence when attempting to correlate the shape of the vortex to sea ice anomalies, with the conclusions managing to be both tentative and brash at times. The findings are interesting, but not yet robust enough. Furthermore, the manuscript is sparsely sourced in many places, putting the onus on the reader to research much of the context and background. It is unclear why so much material is in the Supplementary Material compared to the 4 Figures in the manuscript proper, but the text of the manuscript could easily be reduced substantially to make more room for some of this additional material. Several parts of the results and discussion sections felt a bit disjointed and awkward and could be shortened. The general structure is adequate, but much of the manuscript would benefit from consideration as to whether the flow and phrasing is optimal, a check for missing words and duplication, as well as a sweep for clarity and conciseness. There is considerable potential for this work, and I therefore recommend it for reconsideration after major revisions, with the hope that the authors will find my following comments useful.
* * *
We thank the reviewer for her/his helpful comments and constructive remarks. We agree with the reviewer that interesting insights have emerged from the investigation that we carried out on the quasi-simultaneous evolution of sea ice cover, the tropospheric westerlies, and the polar-front jet stream. As pointed out by the reviewer, the main concept of our manuscript is a thought-provoking and valuable angle of exploration.

The reviewer argues that our conclusions are "both tentative and brash at times". It is an interesting point, and we thank the reviewer for raising it. Certainly, there are some aspects of the study that are more conclusive since they are built upon previously published results. For example, there is prior convincing evidence regarding the stratosphere-troposphere coupling. However, there are other aspects of our study that we have to be cautious about, since we are challenging a commonly accepted hypothesis. For example, the dynamics of the polar vortex is often assumed not to be consequential for the Antarctic sea ice extent. In our manuscript we challenge this hypothesis with evidence that strongly suggests that the stratospheric polar vortex dynamics do play a role in the year-to-year changes of the Antarctic sea ice cover. Although our manuscript will hardly close the case, we do expect that it will trigger the cross-disciplinary work required to further explore the effects of the polar vortex dynamics on the Antarctic sea ice extent. In the revised version of our paper, we have done our best to appropriately scale the emphasis of our conclusions.

We also agree with the reviewer on the significance of some of the information currently shown as "Supplementary Material". We have incorporated his/her suggestions in this regard (two new figures are now included in the main text). As further discussed below, we believe that the clarity and conciseness of our manuscript have greatly improved by the reviewers' remarks and suggestions.
* * *
**General Suggestions of Reviewer 1**

Check that the tenses of consecutive sentences is consistent (e.g. line 80-85). Section 2 (Data and Methods) is too long and wordy. It could easily be condensed down by stating at the top that 'daily estimates' of all data are used, and the reference period from which mean and anomalies are calculated

(except in the case of ozone), and then just stating where each dataset is from, with careful consideration of the most efficient presentation of any remaining information.

Figure captions tended to be too long and with too much detail. They could easily be shortened and tightened up. The submission policies for TC state that, as long as any abbreviations (e.g. datasets used) are already defined in the text, they do not need to be explicitly re-defined in the caption. After a concise description of the actual plot, any extra information would be better placed in the introduction and data/method sections – and I don't think necessary to include the programming package used to create each Figure at the end of each Figure. Perhaps an acknowledgement is a better option.

According to the suggestions of the reviewer, in the revised version of our manuscript, we have:

- Checked the consistency of the tenses of consecutive sentences.

- Shortened Section 2 (Data and Methods) stating that 'daily estimates' of all data are used, and the reference period from which the mean and anomalies are calculated (except in the case of ozone). As suggested by the reviewer, in the new Section 2 we have just stated where each dataset is from, carefully considering the most efficient presentation of the information. Yet, please note that considering that some readers may be not familiar with the stratospheric circulation, some details needed to be provided.

- Shortened and tightened up Figure captions using abbreviations when possible and reducing the degree of details. After a concise description of the actual plot, any extra information has been placed in the introduction and data/method sections.

**Line-by-Line Suggestions of Reviewer 1**

**Line 87:** It is common practice for sea ice extent to be defined in the data/method section, and thereafter referred to by the abbreviation of SIE in the following sections for efficiency.

**Line 101:** Can any of these variables be abbreviated or denoted using a symbol to be more efficient, since they're used multiple times in this (and other) paragraphs?

**Line 109:** There are almost no references in this section, other than for the data itself. Are these domains from the literature, or defined by the authors? The text states that average east-west wind speed between 45°S-75° is near the peak of the polar jet maximum, but the peak itself could be defined here too (and referenced for readers wanting to know more). The text states 'this is a good measure of the overall temperature of the polar vortex' – how do we know? How much variation occurs in zonal mean zonal temperature across the 60-90° domain? Is there a specific reference here for non-specialists?

**Line 128-131:** There is some redundancy here in highlighting the record minimum – a single (or two shorter) sentence/s could be crafted to incorporate this information more efficiently.

**Line 131:** Attributed by whom? Is there a study to which you are referring here? And why do low annual averages suggest the influence of persistent circulation anomalies of climate modes? Aren't the influences of some of these modes seasonal too?

**Line 136:** The text discusses sector-based domains, but the map only names four locations. If common sectors (ABS, RAS etc) are discussed, it would be helpful to clearly define these sectors both in the text and with lines on the map, clearly stating the place names for all sectors and sub-sectors of note.

**Line 145:** The SAM and La Nina discussion comes out of nowhere here. They are briefly discussed in the introduction, but no references are given explicitly here. In fact, between lines 145-157 could be reorganised and reworded entirely for clarity and flow. I really like Figure S4, and wonder if more could be made of it - perhaps incorporated into the body of the manuscript rather than supplementary material. I note the ASL is discussed more at lines 174-179 and again at 250-252; perhaps this would be a good place to incorporate an additional Figure to robustly illustrate your point.

**Line 159:** This states that the enhanced temporal variability occurs 'around Antarctica' – but is this also reflected in spatial variability?

**Line 156** states that the 2017 record event was due to a circumpolar pattern of ice loss, whereas the 2022 record was due to regionally opposing anomalies. Is spatial variability also enhancing over this period, or staying roughly the same?

**Line 164:** Firstly, this line is a direct repeat of lines 43-45. Also, the reference is Doddridge & Marshall, 2017 not Doddridge et al. 2017. Furthermore, the aforementioned reference states that the effect of wind anomalies on sea ice in 2017 was due to the negative SAM in 2016/17 summer, but so far in the manuscript the only discussion has been of persistently positive SAM anomalies. Figure S6 only shows September SAM index, where this negative polarity does not appear. Also, both these references discuss the effect of wind anomalies on the ocean and the combined effect on sea ice, whereas I do not see discussion of these important coupled processes in this manuscript.

**Line 166:** Are the strengthened westerly winds really the cause of the positive SAM?

**Line 168:** Does this hold true only for September? Is there a similar ranking of SAM anomalies for August, or October?

**Line 189-190:** Reconsider routine use of the word 'favor' throughout the manuscript. Reworking this paragraph to include a brief, simple description of why the same driver has opposing seasonal effects would be more impactful.

**Line 222-224:** I'm not enamoured by the argument that, of 42 years, 6 of the highest 15 occurred in the past 10 years. Why choose the top 15 highest? A qualitative look at Figure S10b tells me that 3 of those years were in the 80s, 5 were in the 90s, a single year was in the 2000s, and 6 occurred post-2015. If you only chose the top 10, the 2000s and beyond would barely feature compared to the 90s. I think there is probably a better way to present your argument, as this feels less than robust.

**Line 224-231:** I'm having trouble following the flow here. The strength of the jet since the mid-2010s is remarkable because of the healing ozone hole, but it is unlikely that ozone depletion provided a considerable contribution to it? In fact, the entire paragraph 230-240 is difficult for me to follow. Consider rewording for clarity and flow.

**Line 256:** In general, yes, compatible. But there are notable differences, which are not discussed. For the 2022 case, the Bellingshausen and Amundsen seas region are a stark contrast, and even the Ross Sea region to a lesser extent is not closely matched. In the 2017 case, the anomalies match more closely. Is it possible to condense these plots down? Does (b) really need shading? Perhaps the mean vortex line could be overlaid in a different colour to the mean extent line on the ice concentration plots, and stippling instead of shading for the vortex pattern? It would make it much easier to compare them rather than shifting constantly between panels.

**Line 279-281:** This feels overly pithy, and would benefit from rewording for clarification and flow. It gives a sort of 'shrug' impression to the end of the results section, which is oddly deprecating.

**Line 291:** I've seen this 'all-time records' a couple of times in the manuscript, and I think it would be better to frame this as according to the satellite data record, which only spans 43 years.

**Line 301-302:** Why are these underlined? It's distracting. If worried that the reader will not pick up on the difference between the two words, perhaps word it differently so it is clearer.

**Line 303:** I'm not sure about this. Technically they aren't remarkably nor consistently strong compared to previous decades, at least according to Figure S10 – the value for 2021 is the highest but 1982 was almost as high, and 2016, 2018 and 2020 rank well under the 5 ranking years from the 90s. 2017 doesn't even make the table. Compared to the 2000s, yes, the values are remarkably stronger, but is that what is meant here?

**Lines 321-331:** Requires editing and reworking – several adjoining words are missing and others are superfluous. The flow is difficult, and the conclusion seems hesitant. By contrast, the conclusion of lines 335-7 seems overly confident based on the evidence presented, particularly with no discussion of the effects of SST nor much of a mention of spatially heterogeneous or competing drivers.

According to the suggestions of the reviewer, in the revised version of our manuscript, we introduced the following changes:

- Line 87 (see line 34 in the revised version): Defined the sea ice extent in the data/method section and use thereafter the abbreviation of SIE.

- Line 101 (see line 97 in the revised version): Used abbreviations in the case of the sea-level pressure (SLP), and the potential vorticity (PV).
- Line 109 (see line 104 in the revised version): Clarified that using these domains is a common practice and provide references supporting our statements regarding the peak of the polar jet maximum and the overall temperature of the polar vortex.
- Line 128-131 (see line 122-125 in the revised version): Reworded these lines avoiding redundancies.
- Line 131 (see line 126 and lines 140-143 in the revised version): Cited prior efforts that have studied first, the effect of deep storms on annual SIE minima, and second, the influence of climate modes (operating on seasonal or annual timescales) on the annual SIE mean.
- Line 136: Identified the sea sectors discussed in the text in Fig. 1b (Fig. 1c in the revised version).
- Line 145 (see lines 140-143 in the revised version): Reorganized and reworded lines 145-157 for clarity and flow and provided references supporting our discussion involving SAM and La Nina.
- Incorporated Figure S4 into the body of the manuscript (see new Fig. 2). As noted by the reviewer, this Figure can support the point we make in lines 174-179 and 250-252 (see lines 149 and 246 and lines 140-143 in the revised version).
- Line 159 (see line 155 in the revised version): Clarified that the text in this line refers to the temporal variability only. However, we pointed out in the prior paragraph that the 2017 and 2022 SIE anomalies exhibited considerably different regional patterns. Although the latter suggests an enhanced spatial variability in some Antarctic sectors, please note that the relatively short period of enhanced temporal variability (since 2016) makes it difficult to identify sectors of enhanced spatial difference.
- Line 156 (see line 151 in the revised version): Clarified that, although the 2017 and 2022 SIE anomalies concurred with persistently positive SAM values, they exhibited considerably different regional patterns.
- Line 164 (see line 161 in the revised version): Avoided needless repetitions and corrected reference (from Doddridge et al. 2017 to Doddridge & Marshall, 2017).
- Clarified that while Doddridge & Marshall (2017) studied the effects of SAM anomalies in the austral summer, we focused on the effects of SAM anomalies in the austral spring. We agree with the reviewer that both references now in line 161 (Eayrs et al., 2021; Doddridge & Marshall, 2017) address points beyond the scope of our paper. However, both references underline the influence of the SAM and the westerly winds on the SIE, which is one of the main points made in our manuscript.
- Line 166 (see line 164 in the revised version): Rephrased the text avoiding any suggestion of causality.
- Line 168 (see line 165-166 in the revised version): Confirmed that the SAM index often exhibited strong positive values both in winter months (from 2004 to 2016) and in spring months (from 2016 to 2022).
- Line 189-190 (see line 167 in the revised version): Avoided the use of the word 'favor' and reworked this paragraph to include a brief description of why the same driver has opposing seasonal effects.
- Line 222-224: Eliminated the unnecessary argument regarding the strength of the jet. We agree with the reviewer that it is not particularly compelling.
- Line 224-231: Rephrased the paragraph (see now lines 225-235) for clarity. According to the comment of the reviewer, we have clarified that although the ozone depletion provided a considerable contribution to strengthening the jet stream in the past, the ozone healing observed since the mid-2010s has likely negated that contribution.
- Line 256: Discussed the notable regional differences shown in Fig. 5a (lower panel versus upper panel) between springtime sea ice anomalies in 2022 and 2017 (see first paragraph of section 3.4). Please note that the differences underlined by the reviewer in the Bellingshausen-Amundsen and the Ross Sea sectors are compatible with anomalies in the Amundsen Sea Low (deeper in 2022 than in 2017; see new Fig. 2), which are extensively discussed in section 3.1.

- Explored different variants for Fig. 5 following the helpful suggestions of the reviewer. We have tried to make it easier to compare plots a) and b) by identifying common sectors (Amundsen Sea, Ross Sea, etc.) in the plots. We also included a new set of plots (see new Fig. 5c) showing the low-level air temperature anomalies. We believe that this new set of plots allows us to better link stratospheric anomalies with heat advection and in turn SIE anomalies.
- Line 279-281 (see lines 275-277 in the revised version): Reworded these lines for clarification. We appreciate the comment of the reviewer. It is of course not our intention to be self-deprecating, but we believe that it is important to be transparent regarding the limitations of the available evidence.
- Line 291 (see line 284 in the revised version): Avoided the repetitive use of the expression 'all-time records'. According to the suggestions of the reviewer, we now discuss about "satellite-era records".
- Line 301-302: Eliminated underlined text and make the text clearer.
- Line 303 (see line 299 in the revised version): Linked the text in line 303 with Fig. 3a, which shows that the significant losses in the sea ice extent (that started to occur around Antarctica in 2016) coincided with remarkably strong tropospheric westerly winds in September.
- Lines 321-331 (see lines 316-323 in the revised version): Edited these lines paying special attention to the adjoining words.
- Better calibrated the emphasis of our conclusions and explicitly stated some of the limitations of this work, such as the effects of spatial heterogeneity or competing drivers, like the SST (see lines 333-339 in the revised version).
* * *
**Suggestion of Reviewer 1 (Figures)**
**Figure 1a:** I found it difficult to make sense of the arrows here. Why is the minimum arrow pointing near to the sea ice maximum, while the maximum arrow pointing towards the summer? I think the caption is a better place for this detail, as the presence and positioning of the arrows detracts from its readability.
In addition, I wonder if higher density is warranted on the X axis, and perhaps gridlines to show the start of each month rather than just a rough indication of each 3-month period. The text discusses late February and early September, but only shows a little marker for 1-Jan and 1-Oct. I'm not a big fan of the stretched Y axis either; I guess it shows the minimum more clearly but loses a lot of detail in the early September period which is also discussed in the text.
**Figure 3b:** Again, why not remove the arrow and simply state the 30-day running mean beneath the reference period in the legend for this part of the Figure? It is already stated in the caption that the bold line is the 30-day centered moving mean.
**Figure S4:** This seems an important and clear difference between the two years, though wind vectors overlaid on both plots to show anomalous wind directions resulting from the pressure anomalies would be a welcome addition.
* * *
According to the suggestions of the reviewer, in the revised version of our manuscript, we have

- Eliminated arrows in Fig. 1a and used the caption for details.
- Used a linear scale in Y axis and included gridlines to show the start of each month in Fig 1a.
- Eliminated arrows in Fig. 3b and used the caption for details (see new Fig. 4b).
- Included isobars (which indirectly indicate the predominant wind direction) in Fig. S4. Note that, in the revised version, Fig. S4 was moved to the main text (see new Fig. 2).
- We also included a new set of plots (see new Fig. 5c) showing the low-level air temperature anomalies. We believe that this new set of plots allows us to better link stratospheric anomalies with heat advection and in turn SIE anomalies.

Finally, please note that the revised version of our manuscript now also mentions the record low annual minimum set in February 2023. We thank the reviewer once again for his/her helpful suggestions and

constructive remarks. We believe these revisions have greatly improve the clarity and conciseness of our manuscript. As explained above, the revised version of our manuscript has been modified accordingly.

Best regards,

The authors.

**Final author comments (ACs); manuscript tc-2023-59**

Comments of Anonymous Referee #2
* * *
**General Suggestions of Reviewer 2**
This manuscript tackles the perplexing behavior of Antarctic sea ice for 2013-2022, especially the remarkable retreat during 2016-2022. Following the slow but steady expansion of sea ice extent since the start of the satellite record in 1979, a period of greatly enhanced variability commenced in 2013. Thus, there is high scientific merit in seeking to understand this unexpected behavior. That being said, I found this manuscript challenging to understand even after reading it twice. The broad range of material tackled contributes to my understanding challenge, likely due to my unfamiliarity with some aspects of sea ice behavior. Also in a very unusual departure, the analysis draws substantially on the supplementary material; the authors should consider moving the most important figures into the main manuscript. Mostly I would classify the specific comments below as a major-minor revision.
* * *
We thank the reviewer for her/his helpful comments and constructive remarks. We agree with the reviewer that there is high scientific merit in seeking to understand the recent record-breaking Antarctic sea ice anomalies.

We completely agree with the reviewer that the manuscript covers a "broad range of material". Certainly, studying the response of the sea ice extent around Antarctica to changes in the polar vortex strength is complex, which can make some parts of the text challenging for some readers. However, we believe that the reviewers' remarks and suggestions have allowed us to greatly improve the flow and phrasing our manuscript making the reading experience informative and rewarding.

We also agree with the reviewer on the significance of some of the information currently shown as "Supplementary Material". We have incorporated all of his/her suggestions in this regard (two new figures are now included in the main text).
* * *
**Specific Suggestions of Reviewer 2**
Page 4: Why did you use ERA5 for some variables and MERRA2 for others?
Line 122: 10 year (2001-2010) reference period for ozone hole area?
Line 166: "associated with" rather than "the cause of"?
Line 263: How can "cold air to dip northward" lead to "sea ice retreat" in spring 2017? Similarly, the explanation for the sea ice advance in spring 2022 is unconvincing. These comments refer to the Ross Sea and Amundsen Sea region.
The "shapeshifting vortex's effects" (Section 3.4) rely on the stratospheric anomalies to infer the surface temperature advection. You should show the actual surface circulation anomalies from ERA5 to back up your argument relating low-level air temperatures to sea ice anomalies.
Lines 270-272: I don't see this association.
Lines 327-330: You are left with unknown cause(s) of the "early spring strengthening of the polar vortex since the mid 2010s" that you have linked with the retreat of the sea ice cover 2016-2022.
Figure 2b: Are these anomalies statistically significant?
* * *
According to the suggestions of the reviewer, we have:

- Line 87 (see now line 117): Explained why we used both ERA5 and MERRA2 (basically in order to illustrate that both datasets are consistent and that our results do not depend on the used dataset).
- Line 122 (see now line 117): Corrected the typo.
- Line 166 (see now line 164): Rephrased according to the suggestion of the reviewer.
- Line 263 (see now line 257): Rephrased the text in order to clarify that the comments refer to the Ross Sea and Amundsen Sea region. As shown in the new Fig. 5, anomalies in this region allowed

cold air to dip northward in Sep 2022, likely contributing to the sea ice advance in the Amundsen region in spring 2022.

- Added a new figure (see new Fig. 5c) showing low-level air temperature anomalies. This new figure allowed us to further analyze the effects of the shapeshifting vortex on the heat advection and in turn on the sea ice cover.
- Lines 270-272: Rephrased the unclear association described in text Lines 270-272.
- Lines 327-330 (see now lines 334-340): Expanded upon the limitation of our work and the need for further research on the influence of other competing drivers.
- Figure 2b (see new Fig. 3b): Included stippling indicating regions where the wind speed anomalies can be considered to be significant according to the Welch's t-test ($p \le 0.05$). Please note that in the new version of paper this plot corresponds to Fig. 3b.

Finally, please note that the revised version of our manuscript now also mentions the record low annual minimum set in February 2023. We thank the reviewer once again for her/his helpful suggestions and constructive remarks that we believe they have greatly improved the clarity of our manuscript. As explained above, the revised version of our manuscript has been modified to address all the points raised by the reviewer.

Best regards,

The authors.

---

## Author Response (AR2)

**Comments of Anonymous Referee #1**
* * *
**General comments of Reviewer 1**

The authors have made substantial improvements to the manuscript, and I therefore recommend this work be accepted following some minor modifications, which I list below.

Some typos still exist within the manuscript – a couple are listed below but a thorough proof-read by the authors is necessary.

There is still too much detail included in the majority of figure captions that should be located within the text instead (or already is, but is repeated in the captions).
* * *
We thank the reviewer once again for their helpful suggestions and constructive remarks.

Following their new suggestions, we have subjected the attached version to a thorough proofread by the authors. We have also made slight modifications to the figure captions in an attempt to improve their clarity and conciseness.
* * *
**Specific suggestions of Reviewer 1**

Lines 19-20: This is a long sentence that would benefit from at least a comma somewhere.
Line 88-9: Technically NSIDC terms their SIC as a percentage (i.e. 0-100, see documentation of NSIDC data) rather than area fraction (0-1), so maybe use 15% here rather than 0.15.
Line 101 and elsewhere in Data/Methods: add an "S" after the degrees symbols for clarity.
Line 122: This makes it look like the record was reached on 21 February in both years, may want to rephrase to make this clearer.
Line 159: This new addition requires some finesse. The first line "This enhanced variability appears to be related to swift changes in the tropospheric westerly winds" is basically a direct repeat of a sentence in the introduction (see lines 45-47) but without the references. Are you intending an "as mentioned above", or is this based on your results. If the latter, how does the enhanced variability "appear" to be related to the westerlies? Your measure of variability spans much further than the SAM average in Figure 3. Figure 1 doesn't show the link either, so this statement either needs a (very) brief explanation and references or to be reworded to more clearly link to whichever of the plots/Supplementary Material shows this to be the case.
Line 198: "index" appears here twice.
Line 208: This sentence needs a reference.
Line 209: Is the feedback loop between the vortex and ozone hole, or the other way around? The way the following sentence is written, the loop commences from the ozone – is that what you intend to say? This begs some clarification.
Line 246: "comparable", not "camparable".
* * *
We have made modifications to the text in accordance with each of the minor suggestions mentioned above. In the specific cases of lines 159, 208, and 209, we have slightly modified the text and included additional references that further explore some of the interesting points raised by the reviewer.
* * *
**Additional Suggestions of Reviewer 1**

Figure 4: It would be far easier for the reader to understand the links between the text and the figure if the axis titles were the same as what is being described in the text. Instead of "45-75°S Zonal Wind Anomaly (m/s)" etc, consider simply titling the axes as "stratospheric jet stream" and "tropospheric westerly winds", with the more technical description in data/methods or within the relevant results section in the body of the manuscript.

Figure 5: This is a welcome modification, but I'm curious why you don't just show the Spring 2022/2017 anomalies, relative to the 1981-2010 mean. Wouldn't that be easier to interpret relative to the temperature anomalies shown in (c)? This is done in Figure S3 for 2022, but is stuck in the supplementary material.

According to these suggestions, we have slightly modified the captions of Figs. 4 and 5. In addition and as requested by the reviewer, we have modified Fig. S3 adding the sea ice anomalies for 2017.

We thank the reviewer once again for his/her helpful suggestions and constructive remarks. We believe these revisions have greatly improve our manuscript.

Best regards,

The authors.

**Final author comments (ACs); manuscript tc-2023-59**

**Comments of Referee #3**

> **General Suggestions of Reviewer 3**
> The authors study the links between sea ice variations and general atmospheric circulation, in particular what happens in the stratosphere. I find the paper rather well constructed with more than useful revisions. The work between the two versions seems to me to respond perfectly to the reviewers' remarks.
>
> The introduction is a bit long, but clear and complete. It gives readers from different backgrounds a good understanding of how atmospheric circulation works and what we (dont') know today. Numerous additions in the latest version answer questions I had while reading (e.g. why use ERA5 and MERRA2, mentioning the opposite role of SAM according to season explicitly). I also found very interesting the last paragraph before the conclusion, which puts into perspective what we are currently observing over a longer period. The authors are particularly honest about what they have found and what we can conclude from it.

We thank the reviewer for his helpful comments and constructive remarks. We agree with the reviewer that previous revisions greatly improved the clarity of our manuscript. As highlighted by the reviewer, we have attempted to be transparent regarding both the scope and limitations of our work.

> **Specific Suggestions of Reviewer 3**
> My only minor/technical concern is about Figure5. The caption says there is a link between large-scale stratospheric flow and sea ice variation, but shows 700hPa temperature which is the troposphere. Following the paper, I have no doubt there is a link but maybe this level is not the more adequate with the caption.

According to these suggestions, we have slightly modified the caption clarifying that Fig. 5b shows anomalies in the *stratospheric* flow, while Fig. 5c shows anomalies in the *tropospheric* air temperature.

We thank the reviewer once again for his helpful suggestions and constructive remarks.

Best regards,

The authors.